# Research on Rapid and Low-Cost Spectral Device for the Estimation of the Quality Attributes of Tea Tree Leaves

**DOI:** 10.3390/s23020571

**Published:** 2023-01-04

**Authors:** Jinghua Wang, Xiang Li, Wancheng Wang, Fan Wang, Quancheng Liu, Lei Yan

**Affiliations:** 1School of Technology, Beijing Forestry University, No. 35 Tsinghua East Road, Beijing 100083, China; 2Bureau of Ecology and Environment of Hanting District, No. 1507 Fenghua Road, Weifang 261100, China; 3Information Technology Research Center, Beijing Academy of Agriculture and Forestry Sciences, Beijing 100097, China

**Keywords:** multi-channel spectral sensor, tea leaf ingredients, nondestructive detection, sensor design, machine learning, regression model

## Abstract

Tea polyphenols, amino acids, soluble sugars, and other ingredients in fresh tea leaves are the key parameters of tea quality. In this research, a tea leaf ingredient estimation sensor was developed based on a multi-channel spectral sensor. The experiment showed that the device could effectively acquire 700–1000 nm spectral data of tea tree leaves and could display the ingredients of leaf samples in real time through the visual interactive interface. The spectral data of *Fuding* white tea tree leaves acquired by the detection device were used to build an ingredient content prediction model based on the ridge regression model and random forest algorithm. As a result, the prediction model based on the random forest algorithm with better prediction performance was loaded into the ingredient detection device. Verification experiment showed that the root mean square error (RMSE) and determination coefficient (R^2^) in the prediction were, respectively, as follows: moisture content (1.61 and 0.35), free amino acid content (0.16 and 0.79), tea polyphenol content (1.35 and 0.28), sugar content (0.14 and 0.33), nitrogen content (1.15 and 0.91), and chlorophyll content (0.02 and 0.97). As a result, the device can predict some parameters with high accuracy (nitrogen, chlorophyll, free amino acid) but some of them with lower accuracy (moisture, polyphenol, sugar) based on the R^2^ values. The tea leaf ingredient estimation sensor could realize rapid non-destructive detection of key ingredients affecting tea quality, which is conducive to real-time monitoring of the current quality of tea leaves, evaluating the status during tea tree growth, and improving the quality of tea production. The application of this research will be helpful for the automatic management of tea plantations.

## 1. Introduction

Tea (*Camellia sinensis* (L.) *O. Ktze*) is one of the most important economic crops in the world. The world’s top five tea producers are China, India, Kenya, Sri Lanka and Vietnam. Currently, China’s annual production of tea exceeds 2 million tons. Tea is divided into green tea, white tea, yellow tea, red tea, black tea, and oolong according to the differences of variety, production method, and product appearance [1]. As a popular drink, various countries have introduced relevant standards for tea products. However, when fresh tea is purchased for commercial purposes, the quality of fresh tea leaves mostly relies on subjective assessment by practitioners, and there is a lack of systematic detection and evaluation methods. [2]. As an evergreen perennial crop that grows in acidic soil, tea tree leaves are an important embodiment of product value. After being processed by a variety of different techniques, the tea tree leaves can be processed into various tea drinks. Drinking tea has many health benefits: such as anti-oxidation [3], excitation of the central nervous system [4], prevention and treatment of diabetes [5], etc. Tea drinks contain tea polyphenols, free amino acids, soluble sugar, and other ingredients. The difference in the proportion of these ingredients is the essential reason for the difference in the quality of tea products, and carbon, nitrogen and carbon–nitrogen ratio are the important basis for the formation of key physical and chemical ingredients of tea. Therefore, detecting the ingredient information of tea tree leaves has great significance for accurately acquiring the growth status of tea trees, ensuring the quality of tea products, and improving economic value.

The traditional detection methods for tea leaf grading are mainly chemical analysis [6]. Chemical analysis detection methods are complicated. Experimenters need to master professional operating skills [7], so it is difficult to popularize and apply. At the same time, analyzing the ingredients of tea tree leaves through chemical analysis is a destructive detection method [8].

At present, the non-destructive detecting methods of tea tree leaf ingredients include machine vision detecting methods [9,10], chemical sensor detecting methods [11], spectral technology detecting methods, etc. Among them, the detection method based on spectral technology has the advantages of fast detection speed, high detection efficiency, and no need for pretreatment. It has a good application prospect in the detection of fresh tea leaf ingredients. At present, many researchers have carried out extensive research. These studies explored the feasibility of optical detection methods [12] in predicting the ingredients (tea polyphenols, theanine, etc.) in tea. Some reports proved that spectral technology had a high application value in the real-time detection of tea ingredient information [13]. Luo et al. [14] proposed an efficient and high-precision prediction model for the content of tea polyphenols based on near-infrared spectroscopy. Many researchers have also used spectral technology to detect and predict the moisture content in tea tree leaves [15,16,17]. Liu et al. [18] studied the moisture content changes during green tea processing based on near-infrared spectroscopy combined with machine vision technology. Wang et al. [19] combined hyperspectral techniques with chemical determination methods to propose a prediction model for moisture, nitrogen, crude fiber content, quality index value, and other ingredients of different varieties of tea. Wang et al. [20] proposed a non-destructive detection method of macro elements (phosphorus and potassium) in tea leaves based on spectral technology. The spectral analysis experiments in the above research were mainly carried out in the laboratory environment. Spectral data of samples were collected by a benchtop near-infrared spectrometer or a hyperspectral imager. These instruments are bulky and costly, which is not conducive to practical application in tea plantations. Therefore, it is necessary to develop a convenient and accurate real-time detection device for tea tree leaf ingredient information detection.

With the development of optical sensors, many miniaturized non-destructive detection devices for agricultural products based on optical sensors have been manufactured. Some researchers have manufactured a tea tree leaf quality evaluation device based on a miniature spectrometer [21,22]. Wang et al. [23] developed a portable micro near-infrared spectrometer to detect the content of catechin and caffeine in black tea samples. Ren et al. [24] identified the quality of *Dianhong* red tea based on the method of near infrared spectroscopy and stoichiometry. However, the existing devices of this kind are mostly aimed at the detection of single-ingredient parameters. There is still a lack of detection device for comprehensive ingredients and parameters of tea tree leaves. Based on the above problems, this research designed a detection device for fresh tea leaf ingredient information based on a multi-channel spectral sensor. At the same time, spectral real-time analysis software with a random forest prediction model was loaded to realize rapid and non-destructive detection of water content, tea polyphenols, total free amino acids, soluble sugar content, total nitrogen content, and chlorophyll content of fresh tea leaves. At the same time, a spectral real-time analysis software equipped with a random forest prediction model was written. The software realizes rapid and non-destructive detection of ingredients of tea tree leaves. It was expected to provide an effective technical means of real-time monitoring of the current quality of tea leaves, evaluating the status during tea tree growth, and improving the quality of tea production.

## 2. Materials

When light hits the surface of tea tree leaves, the difference in texture and color of the leaf surface will affect the reflected light. Different chemical bonds and functional groups inside tea tree leaves will also respond to different information in specific spectral bands. Therefore, various ingredients of tea tree leaves can be detected through the analysis and inversion of reflectance spectral information. Based on the above principles, this research developed a detection device for ingredients of detached tea leaves based on spectral technology.

The development of tea tree leaf ingredients information detection device mainly included hardware system design and software design. The hardware system design included sample chamber, spectral acquisition, power supply, cooling, control, and display unit design. The software was developed on the Linux system using the Qt software. Functions such as multi-channel spectral sensor setting, spectral acquisition, spectral preprocessing, result prediction, result display, and storage could be realized. The prototype was shown in Figure 1.

### 2.1. Hardware System Design

#### 2.1.1. Sample Chamber Unit

The environment for acquiring spectral information of tea tree leaves needed to meet two principles: The first principle was that the incident light spectrum must cover the target reflected light spectrum so that the spectral sensor could receive the reflected light of the target band; the second principle was that the collection environment needed to build a dark room to avoid the leakage of light sources and the interference of external ambient light. The sample room unit was designed based on the above principles, and the layout of the sample room is shown in Figure 2. The sample chamber unit size is 15 cm × 10 cm. The sample chamber was partially made of resin material by 3D printing in order to reduce the light reflection phenomenon inside the sample chamber. In the cause of reducing the influence of external ambient light on the test results, the inside of the chamber was sprayed with black matte material to construct a dark room.

A multi-channel spectral sensor and light source were installed above the chamber. A Philips G4 tungsten halogen lamp bead was selected as the light source. Its power was 45 W, the rated voltage was 12 V, and the lamp head width was 4 mm.

A groove was in which to place the diffuse reflectance standard plate and the tea leaf sample was designed at the bottom of the chamber. *Jingyi* JY-WS1 diffuse reflection standard white board was selected as the diffuse reflection standard board. The standard whiteboard diameter was 50 mm and was made of PTFE (polytetrafluoroethylene) material. Through testing with the American Ocean Optical Reflectivity Test System, the results showed that the reflectance of the standard whiteboard was 99.3% and that the uniformity was ±0.04% (environment temperature of 25 °C, relative humidity of 40%), which met the design requirements.

#### 2.1.2. Spectral Acquisition Unit

The spectral acquisition unit was designed based on the Ocean Optics PixelSensor multi-channel spectral sensor. The sensor integrated eight-band simultaneous sensing photodiodes onto a 9 mm × 9 mm optic. The PixelSensor chip-level optical filter technology equipped with the sensor directly splits the spectrum into eight independent color bands, and the background light of other bands was well suppressed, thereby improving the contrast and sensitivity. The sensor photodiode performance parameters were shown in Table 1. The sensor integrated OEM circuit boards that supported USB 2.0 communication.

#### 2.1.3. Power Supply Unit

To ensure that the device could work for more than two hours in a tea plantation, the device used a total of twelve 18650 lithium batteries connected in three series and four parallel to form a power supply with an output voltage of 11.1 V and a battery capacity of 12 Ah. During the working process, the output voltage of the battery would decrease with voltage fluctuation. This would cause unstable brightness of the light source, which would affect the prediction performance. Therefore, two voltage-stabilizing modules were added to the circuit to realize the smooth operation of the device, as shown in Figure 3.

#### 2.1.4. Cooling Unit

The light source and multi-channel spectral sensor were sensitive to temperature. Tungsten-halogen lamps generated abundant heat during work. Since the internal space of the device was limited and the heat was difficult to dissipate, a cooling unit was designed to maintain a constant temperature inside the device. The DC fan reduced the heat by exchanging the air inside the device with the air outside. The air volume was an important index to measure the cooling capacity of the fan. Excessive air volume would increase the power consumption burden of device. Therefore, the number of fans needed to be set according to the calorific value. The calculation method of the total heat exhausted by a single fan was shown in Equation (1).
(1)$$H=C_{P}\times \left[\left(Q/60\right)\times \rho \right]\times \Delta T_{c}\times \eta$$
where H represents the total heat exhausted by the fan (unit: J); Q represents the air volume of the fan (unit: CFM). η represents the heat dissipation efficiency of the fan, generally taken as 60% (unit: %); ∆T_c_ represents the allowable temperature rise of the container (unit: °C); ρ represents the air density (unit: kg/m^3^); C_P_ represents the specific heat capacity (unit: J/(kg·°C)).

The power of the light source was 45 W. The total power of the multi-channel spectral sensor, Raspberry Pi, and touch screen assembly was 15 W, and the battery was 40 W. The total power of the device was about 100 W. Required air volume Q was shown in Equation (2).
(2)$$Q=\frac{2.2P}{\Delta T_{c}}=5.50CFM$$

This device used an aluminum alloy shell double fan cooling module with an air volume of 6.0 cubic feet per minute (CFM).

#### 2.1.5. Control and Display Unit

The control and display unit was designed based on RaspberryPi4 4B, which was equipped with a BCM2711B0 CPU based on ARM architecture and Broadcom Video Core VI GPU. Th computing capabilities were capable of running the spectral data analysis model, which can realize real-time detection. Two USB3.0 and two USB2.0 interfaces were reserved on the RaspberryPi4 4B, which could communicate directly with the PixelSensor multi-channel spectral sensor through the USB interface to realize data transmission and control. Two HDMI ports were reserved for direct connection with the touch screen.

### 2.2. Software Design

Good interface design could not only give users a good experience but also increase the efficiency of using the software. Visual interactive interface was designed as shown in Figure 4.

The software operation process as follows is shown in Figure 5.

1. Open the software and select a leaf ingredient for detecting (moisture content, free amino acid content, tea polyphenol content, sugar content, nitrogen content, or chlorophyll content);

2. The software loads prediction model according to the ingredient’s parameter selection;

3. Turn on the light source, place the diffuse reflection standard board into the spectral collection area of the sample chamber, and click “Diffuse Reflectance Correction”;

4. Open the side panel of the sample chamber, take out the standard board, and put in the leaf sample. Close the side panel and click “detection” button in the interface to acquire the spectrum;

5. The software automatically performs reflectance calculation, preprocessing, and output of prediction results;

6. The software saves the spectral data and prediction results in a .txt file.

### 2.3. Debugging and Testing

The Raspberry Pi 4B was connected with MIPI DSI capacitive touch display, multi-channel spectral sensor, etc., to the corresponding interface with the program loaded. The multi-channel spectral sensor was placed under different light intensity environments, and spectral data acquired by the device were observed on the MIPI DSI screen to ensure the effectiveness of the collection of spectral information. The integration time and sampling speed in the debugging and testing experiment are shown in Table 2.

## 3. Methods

### 3.1. Spectral Acquisition of Tea Tree Leaves

This experiment was carried out in a tea plantation in Yongchuan District, Chongqing, China. The experimental subject was *Fuding* white tea trees, as shown in Figure 6. Comprehensively considering the quantity of samples required for the chemical experiments, an abundance of tea tree leaves was selected from two sample plots in the tea plantation. In total, 33 *Fuding* white tea trees were selected in each sample plot for sufficient sampling. During the sampling process, fresh leaves were picked from the selected tea tree canopy in accordance with the principle of uniformity and randomness. Tea tree leaves were selected from each tea tree, respectively. A total of 264 samples were obtained, each with about 200 g of tea tree leaves.

### 3.2. Chemical Experiments for Ingredient Content Determination

After the spectrum acquisition is completed, the samples of tea tree leaves were put into the fresh keeping bags for marking and stored at low temperature in a portable incubator. The ingredients content of samples was measured in the laboratory.

The moisture content ($L_{\mathrm{water}}$) of the samples was measured according to Equation (3):(3)$$L_{\mathrm{water}}=\frac{M_{1}-M_{2}}{M_{1}}\times 100\%$$
where *M*_1_ represents the mass of the sample before drying (unit: g); *M*_2_ represents the mass of the sample after drying (unit: g).

The free amino acid content (Y) was measured by ninhydrin method according to Equation (4):(4)$$Y=\frac{\left(\frac{C}{1000}\right)\times \left(\frac{V_{1}}{V_{2}}\right)}{m\times w}\times 100\%$$
where *V*_1_ represents the total amount of tested liquid (unit: mL); *V*_2_ represents the amount of test liquid used for mensuration (unit: mL); *m* represents the sample quantity (unit: g); *w* is the dry matter fraction of the sample.

The tea polyphenol content was measured using the Lowenthal–Neubauer method.

According to the Lambert–Beer law, the chlorophyll content (M) was measured via spectrophotometry according to Equation (5):(5)$$M=\frac{\frac{A_{652}}{34.5}\times Amount\;of\;extract}{Mass\;of\;sample}$$
where A_652_ represents the absorbance of the solution at the band of 652 nm; 34.5 represents the specific absorption coefficient of the solution at band 652 nm.

The sugar content was measured using the anthracene reagent method.

The nitrogen content (D) was measured using the Kjeldahl method according to Equation (6):(6)$$D=\frac{c(V_{2}-V_{1})\times 14.01}{m\times 1000}\times 100\%$$
where *c* represents the concentration of sodium hydroxide (NaOH) solution during mensuration (unit: mol/L); *V*_1_ represents the volume of NaOH solution (unit: mL); *V*_2_ represents the volume of *NaOH* solution in blank experiment (unit: mL); 14.01 represents the molar mass value of nitrogen (unit: g/mol); m represents the sample quantity (unit: g).

### 3.3. Modeling Method

#### 3.3.1. Ridge Regression

Ridge regression (Ridge) is a method for estimating the coefficient of multiple regression model with highly correlated independent variables. This theory was first proposed by Hoerl et al. [25]. When the linear regression model has some highly correlated independent variables, ridge regression is a reliable solution to the imprecision of least squares estimation. It avoids the problem of *X^T^X* determinant approaching 0 by artificially introducing the penalty term *kI_p_*, and its principle is shown in Equation (7):(7)$$\beta_{\mathrm{ridge}}=(X^{T}X+kI_{p}){}^{-1}X^{T}y$$
where *β_ridge_* represents the parameter estimation value of ridge regression; *I_P_* represents the identity matrix of order with *X^T^X*; *k* represents the ridge regression coefficient, which is a constant greater than 0 and represents the artificially introduced error.

#### 3.3.2. Random Forest

Random forest (RF) is a supervised machine learning algorithm [26] which is a bagging algorithm based on decision trees. Random forest can better learn the potential relationship between multiple feature dimensions. It has low complexity and strong anti-interference ability, so it is often used to process high-dimension data.

Random forest regression (RFR) is an important application branch of random forest. Random forest regression models, through random sampling and features, combine multiple classification and regression trees (CART). The prediction results are obtained in parallel. Based on different shred variables and descendants, all values of each feature are traversed. The optimal shred variable and descendants are selected according to the impurity of the descendant after shred. The calculation method is shown in Equation (8):(8)$$G\left(x_{i},v_{ij}\right)=\frac{n_{left}}{N_{s}}H\left(X_{left}\right)+\frac{n_{right}}{N_{s}}H\left(X_{right}\right)$$
where $G\left(x_{i},v_{ij}\right)$ represents the unpurity weighted sum of each descendant; $x_{i}$ represents the tangent variables; $v_{ij}$ represents the corresponding shred value of variables; $n_{left}$, $n_{right}$, $N_{s}$ represent the number of left and right descendants training samples and the total number of training samples after segmentation, respectively. $X_{left}$, $X_{right}$ represent training sample sets of left and right descendants, respectively. H(X) represents the descendant impurity measure function.

#### 3.3.3. Model Training

The spectral technology analyzed the material content based on the fingerprint map of the frequency absorption from the hydrogen group (including C-H, N-H, O-H and S-H) expansion and vibration in different spectral bands. The concentration of the hydrogen group was positively correlated with the absorption intensity. Therefore, ingredients composed of the hydrogen group could be described by the spectral characteristics, and the correlation model of the reflected spectrum and ingredients content was established by combining the chemical measurement.

The combination of tea tree leaf spectral data and ingredients information established the tea tree leaves spectrum ingredients database. The ingredient information was set as labels. The database was divided by 70% and 30% into the modeling set and the validation set, respectively. The verification set data was used as the sample data set for the device prototype verification analysis. The modeling set was divided by 70% and 30% into the training set and the test set. The training set was analyzed using the ridge regression model and the random forest model to build the prediction model of tea tree leaves ingredient content. The performances of both models were tested in the test set.

#### 3.3.4. Evaluation Index

The MAE measures the mean absolute error between the predicted value and the true value. The smaller the MAE is, the better the prediction effect is, and its definition is shown in Equation (9):(9)$$MAE=\frac{1}{n}\overset{n}{\underset{i=1}{\sum }}\left|y_{i}-\hat{y_{i}}\right|$$

The root mean square error (RMSE) is used to measure how much error will be generated by the model in the prediction. The smaller RMSE is, the better the prediction effect is, and its definition is shown in Equation (10):(10)$$RMSE=\sqrt{\frac{1}{n}\overset{n}{\underset{i=1}{\sum }}{\left(y_{i}-\hat{y_{i}}\right)}^{2}}$$

Determination coefficient (R^2^) is used to measure whether the prediction error is larger or smaller than the mean reference error. The closer the R^2^ is to 1, the better the prediction effect is. Its definition is shown in Equation (11):(11)$$R^{2}=1-\frac{\sum_{i=1}^{n}{\left(y_{i}-\hat{y_{i}}\right)}^{2}}{\sum_{i=1}^{n}{\left(y_{i}-\bar{y}\right)}^{2}}$$

In Equations (9)–(11), $y_{i}$ represents the predicted value; $\hat{y_{i}}$ represents the true value.

## 4. Results

### 4.1. Results of Measured Ingredients Content

Some basic statistical data regarding the measured tea leaf ingredients are presented in Table 3. The moisture content in all samples of tea tree leaves ranged from 1.37% to 14.2%; the tea polyphenols content ranged from 6.59% to 13.2%; the free amino acid content ranged from 1.6% to 3.04%; the sugar content ranged from 2.05% to 6.66%; the nitrogen content and chlorophyll contents ranged from 2.22% to 4.91% and 0.14% to 0.39%, respectively.

### 4.2. Spectrum Analysis

Due to the high humidity of the air in the tea plantation during the field experiment, some samples of fresh tea tree leaves showed over-exposure phenomena with high reflectivity. The spectral curves of these samples could not represent the material information contained in the samples and it was difficult to correct by mathematical method. Therefore, the original spectral data of tea tree leaves collected were screened to remove abnormal data. The spectral data of 220 samples were obtained after elimination. The spectral curve of tea tree leaves was drawn for each sample, as shown in Figure 7.

As shown in Figure 8, the spectral curve of all the tea tree leaf samples had the same trend. The reflectivity from 700–760 nm was significantly elevated, and the light reflection intensity was higher, but the change was less than that of the 700–760 nm reflectance. In the range of 760–1000 nm, the light reflection intensity was high but the change was small. All spectral curves showed similar troughs between 970 nm and 980 nm, which is considered to be the characteristic absorption peak of water and could be used to estimate moisture content [27,28]. Research showed that the O-H bond in tea polyphenols was stretched and vibrated strongly near the second-order frequency-doubling region, which was reflected in the spectral bands near 1000 nm. These bands could be used to estimate tea polyphenol content. The main constitution of sugar in tea was three soluble sugars: fructose, glucose, and sucrose. The O-H bond (triple frequency) in soluble sugar had strong scattering property near 980 nm and had evident response to the information of chemical constitution. This band could be used to estimate tea sugar content [29]. Due to the absorption of chlorophyll, 726 nm was a reflection peak. In particular, 710 nm was the “red edge” position of chlorophyll. This band could be used to estimate the chlorophyll content. At about 947 nm, the O-H bond of theanine was bent, and the absorption capacity increased in a strong vibration near a level of frequency. In 1000 nm band, the N-H bond of amino acid molecules was stretched, leading to a strong reflection band of the spectrum [30]. In addition, there was a high correlation between the free amino acid and the nitrogen elements in the tea tree leaves, which could be used to estimate free amino acid and nitrogen contents. The relationship between wavelength and leaf ingredients is shown in Table 4.

### 4.3. Prediction Model of Tea Tree Leaf Ingredients Content

The leaf ingredients R-Square of train set and test set (R^2^_c_, R^2^_t_), root mean squared error of train set and test set (RMSE_c_, RMSE_t_) were used to evaluate model accuracy and stability; the results are shown in Table 5.

As shown in Table 3, the prediction ability of random forest model was better than the prediction model based on the ridge regression. Although the more complex network structure of the random forest model brought larger calculation, the performance of RaspberryPi 4b met the operation of random forest model. Therefore, the prediction model based on random forest was selected to load into the tea leaf ingredient estimation sensor.

### 4.4. Verification Analysis for Device

In the analysis and detection of unknown tea tree leaf samples, the corresponding prediction model was used to analyze the spectral data of samples. Then, the ingredient information of tea tree leaves could be detected quickly. Based on the above principles, verification analysis was carried out to test the detection performance of the tea leaf ingredient estimation sensor.

The device was used to collect the spectral data of the tea tree leaves and detect the content of each ingredient in the samples. Moisture content, free amino acid content, tea polyphenol content, sugar content, nitrogen content, and chlorophyll content predicted by the device were recorded and compared to the measured value of ingredient content measured via chemical experiment, as shown in Figure 8.

The predicted value–measured value (real value) fitting curve (P-R curve) of each ingredient was drawn as Figure 9. The mean absolute error (MAE), root mean square error (RMSE), and determination coefficient (R^2^ score) were calculated according to the predicted value of the device and measured value of the chemical experiment, which was used as the standard to evaluate the prediction accuracy of the device, as shown in Table 5.

Table 6 shows that the sensors can predict some parameters with high accuracy (nitrogen, chlorophyll, free amino acid) but some with lower accuracy (moisture, polyphenol, sugar) based on the R^2^ values.

## 5. Discussion

At present, the main methods for the quality detection of tea tree leaves are sensory review and physical/chemical testing. These traditional detection methods are all destructive testing methods, and there are problems, such as serious subjective error and tedious operation, respectively. The existing devices of this kind are mostly aimed at the detection of single ingredient parameters. There is still a lack of detection devices for the comprehensive ingredients and parameters of tea tree leaves [12,31,32,33,34]. Among this research, a device for detecting the ingredient of tea tree leaves was developed based on the spectral technology to realize the rapid non-destructive detection of the key ingredients affecting the quality of tea. It is beneficial to realize the status detection of tea tree during the growth period, evaluate the current quality of tea tree leaves, and improve the quality of tea production made by the tea tree leaves.

According to the experimental results of tea tree leaf ingredient prediction, the prediction ability of random forest regression model was better than that of ridge regression model. This is because the network depth of the random forest model is deeper than that of the ridge regression model. Therefore, the random forest model is more suitable for analyzing and mining the material information contained in the high-dimensional spectral data in this research.

According to the evaluation indexes score in the verification analysis, the tea leaf ingredient estimation sensor had the highest accuracy for the prediction of chlorophyll content in all six ingredients paraments. This is because the device effectively collected the spectral reflection peak and reflection valley of the chlorophyll at 700 nm [35]. Therefore, the significant correlation between the chlorophyll content and the spectrum data can be effectively obtained [36].

The accuracies for the prediction of nitrogen content and free amino acid content in tea leaves were the second- and third-highest. This indicated that the device effectively collected the spectral data of the spectral absorption enhancement region caused by amino acid O-H bond bending and the spectral data of the strong reflection region caused by N-H bond stretching near 1000 nm. Therefore, the spectral data highly correlated with proteins and amino acids were obtained by the device.

In the fitting curve of moisture content prediction results, there were some discrete values in the lower-left and upper-right parts of the fitting regression line, which may be due to the over-aeration phenomenon caused by water mist and small water droplets on some samples of fresh tea leaves collected. This case affected the accuracy of spectral data collected by multi-spectral sensors. There was another possible case that the operation error occurred in the drying weighing experiment, leading to a certain deviation in the predicted results [37]. After removing the discrete values, the mean absolute error (MAE), root mean square error (RMSE) and determination coefficient (R2 score) of water content prediction were 1.00, 1.13, and 0.473 respectively, which significantly improved the prediction accuracy.

The prediction accuracy of tea polyphenol content in the samples was unsatisfactory, which may be due to the correlation of tea polyphenol content indirectly expressed by the relevant spectral information of chlorophyll and water in the band range of 700–1000 nm [38,39]. Catechins are the main component of tea polyphenols. In this band range, the 2-phenyl-3,4-dihydro-2H-chromen structure of catechin has a certain overtone overlap with aldehydes and ARCH functional group of aromatic compounds in the spectral expression [40]. Therefore, it caused a certain confusion in the spectral correlation analysis.

The next research plan is to use spectral sensors with wider spectrum segments and combine correlation analysis method to acquire characteristic spectrum band so as to develop a regression model with better prediction accuracy of tea tree leaf ingredient information. In the future, we will continue to transfer the proposed device and method to other application fields [41,42,43], such as time prediction, signal modeling, and control systems [44,45,46]. Relevant technologies will be studied to expand the application scope of this model in smart agriculture [47] and the food supply chain [48].

## 6. Conclusions

(1) This research developed a tea leaf ingredient leaf ingredient estimation sensor based on a multi-channel spectral sensor. Experiments showed that the device could effectively collect 700–1000 nm spectral data of tea tree leaves and has functions such as black/white correction and ingredient content prediction. Through the visual interactive display, it could collect, display, and save the ingredient information of tea tree leaves in real time.

(2) Using the tea leaf ingredient estimation sensor, spectral data of leaf samples of *Fuding* white tea were collected. The ridge regression model and random forest model were respectively trained to establish each ingredient content prediction model. As a result, the prediction model based on random forest with better prediction effect was loaded into the detection device. R^2^ and RMSE values for the prediction model based on random forest algorithm of moisture content, free amino acid content, tea polyphenol content, sugar content, nitrogen content, and chlorophyll content were 1.607 and 0.35; 0.162 and 0.79; 1.354 and 0.284; 0.138 and 0.334; 1.154 and 0.914; and 0.02 and 0.973, respectively. Results showed that the nitrogen, chlorophyll, and free amino acid content were predicted with highest performance. The moisture and tea polyphenol content were predicted with lower performance. The verified experiment showed that the detection accuracy of tea leaf ingredient estimation sensor loaded with random forest prediction model can meet the requirements of a tea plantation. When farmers sell fresh tea and tea product manufacturers purchase fresh tea, the equipment can be used to comprehensively detect and evaluate the quality of the fresh tea. Compared to the traditional method of subjective grading of fresh tea, this method can improve the accuracy and efficiency of tea quality assessment.

## Figures and Tables

**Figure 1 sensors-23-00571-f001:**
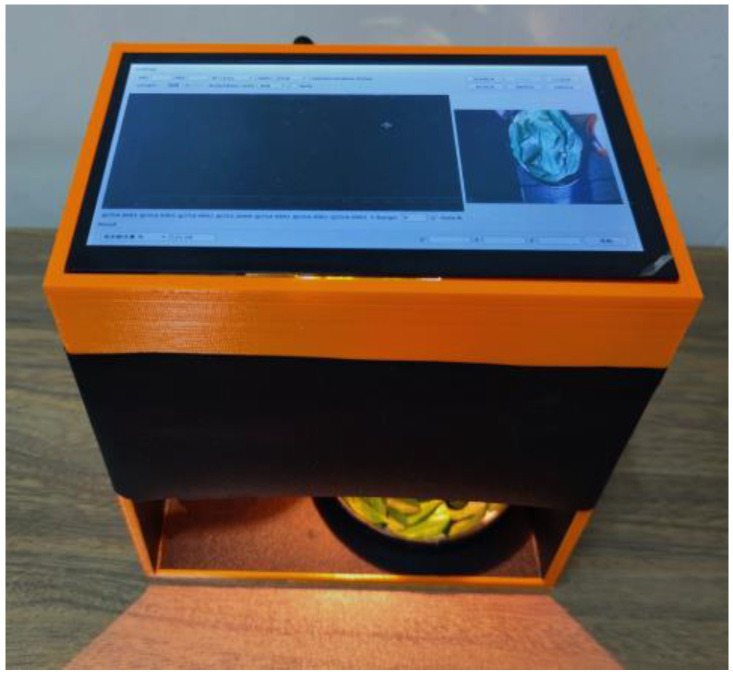
Tea leaf ingredient estimation sensor.

**Figure 2 sensors-23-00571-f002:**
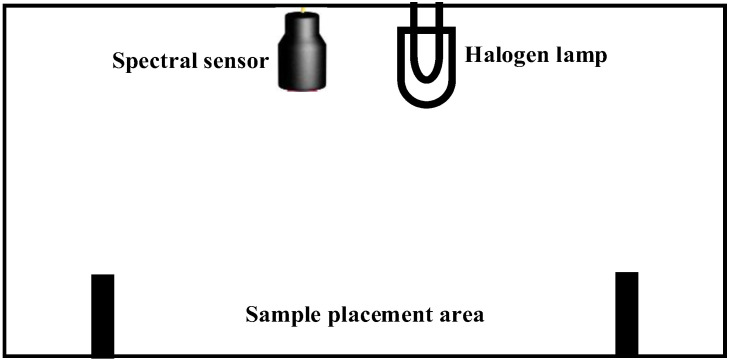
Structure diagram of the sample chamber unit.

**Figure 3 sensors-23-00571-f003:**
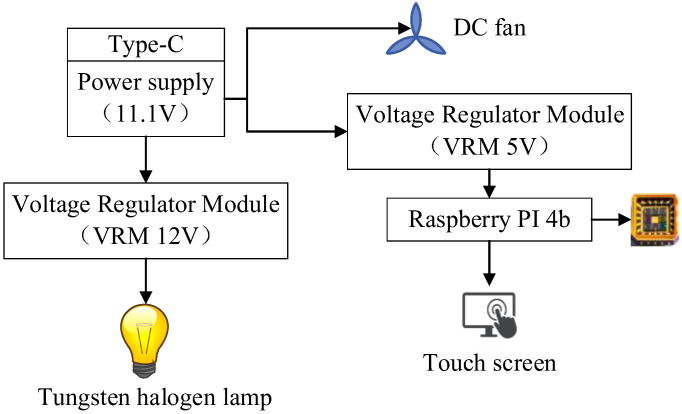
Structure diagram of the power supply unit.

**Figure 4 sensors-23-00571-f004:**
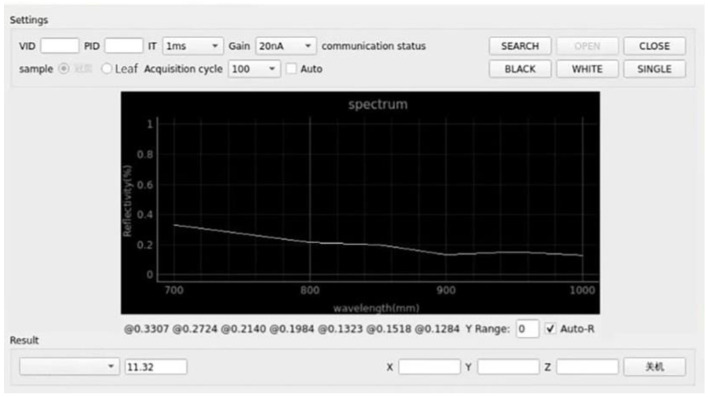
Visual interactive interface.

**Figure 5 sensors-23-00571-f005:**
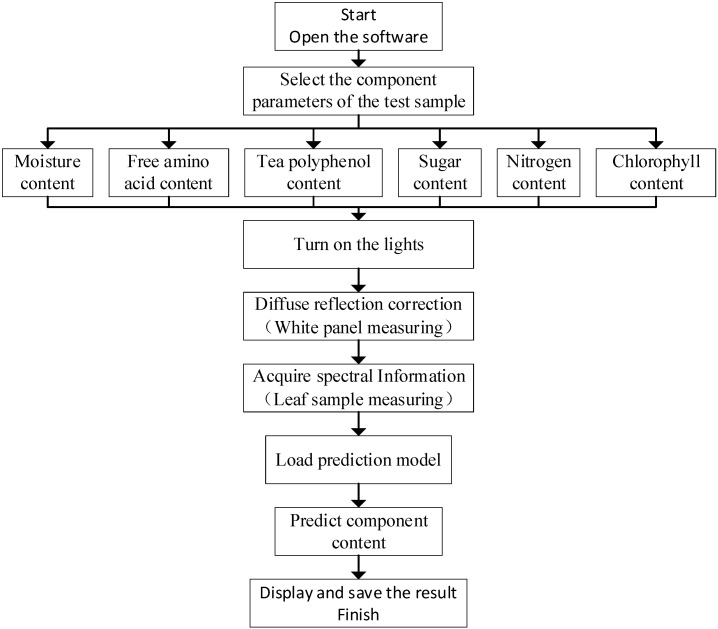
Flow chart of the software operation process.

**Figure 6 sensors-23-00571-f006:**
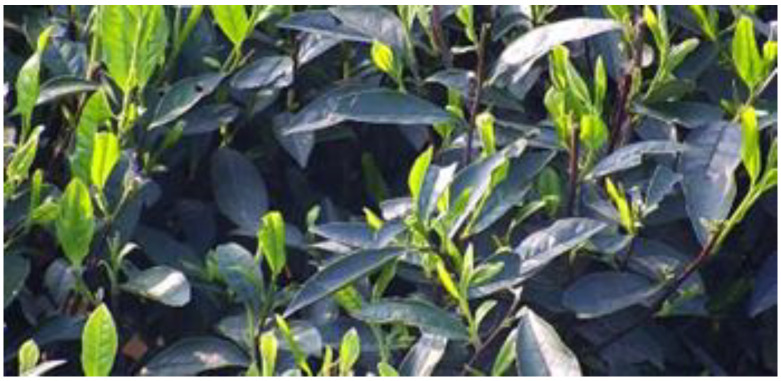
*Fuding* white tea tree.

**Figure 7 sensors-23-00571-f007:**
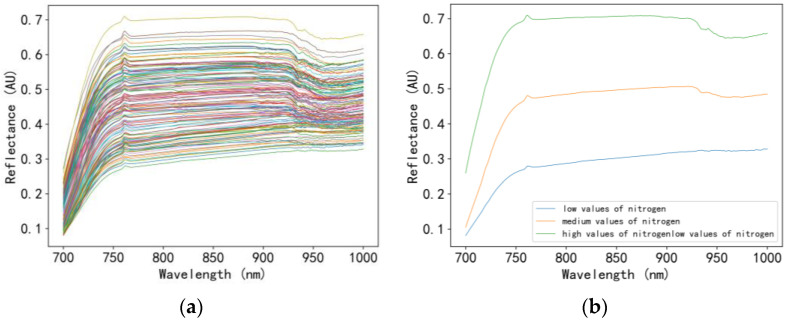
Spectral curve: (**a**) original spectral curve of the tea tree leaf samples; (**b**) mean reflectance curves for high, medium, and low values of nitrogen.

**Figure 8 sensors-23-00571-f008:**
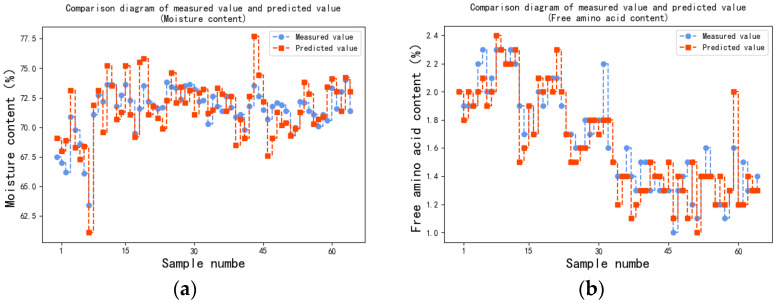
Comparison diagram of the measured and predicted values: (**a**) moisture content; (**b**) free amino acid content; (**c**) tea polyphenol content; (**d**) sugar content; (**e**) nitrogen content; (**f**) chlorophyll content.

**Figure 9 sensors-23-00571-f009:**
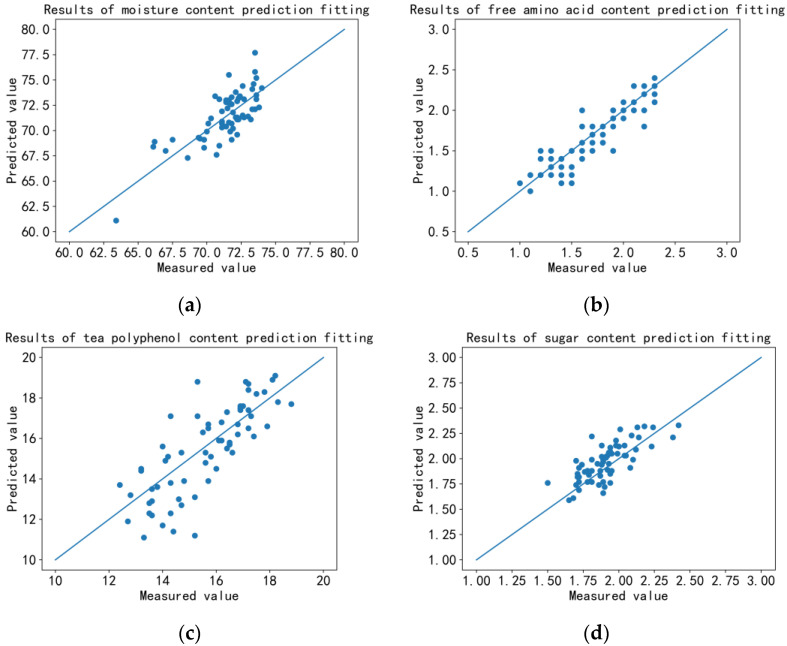
The predicted value–measured value fitting curve: (**a**) moisture content; (**b**) free amino acid content; (**c**) tea polyphenol content; (**d**) sugar content; (**e**) nitrogen content; (**f**) chlorophyll content.

**Table 1 sensors-23-00571-t001:** Ocean Optics PixelSensor Photodiode performance parameters.

Parameters	Value
Dark current	2 nA
Shunt resistor	100 MΩ
Spectral range	700–1000 nm
Response time	6.0 ns

**Table 2 sensors-23-00571-t002:** Result of debugging and testing experiment.

Test Items	Measurement Value
Integration time	1–1024 ms
Sampling speed (storing to RAM)	0.5 ms/time
Data transmission speed	1.3 ms/time

**Table 3 sensors-23-00571-t003:** Some basic statistical data on the measured tea leaf ingredients.

Paraments (Number of Data = 264)	Minimum Value	Maximum Value	Mean Value	Standard Deviation
Moisture content	1.37	14.2	5.06	2.56
Free amino acid content	6.59	13.2	10.21	1.35
Tea polyphenol content	1.6	3.04	2.24	0.3
Sugar content	2.05	6.66	3.58	0.96
Nitrogen content	2.22	4.91	3.96	0.66
Chlorophyll content	0.14	0.39	0.29	0.1

**Table 4 sensors-23-00571-t004:** Relationship between wavelength and leaf ingredients.

	Chlorophyll	Nitrogen and Free Amino Acid	Tea Polyphenol and Water
Wavelengths	700 nm	Near 1000 nm	700–1000 nm

**Table 5 sensors-23-00571-t005:** Evaluation index scores of ridge regression model (RR) and random forest model (RFR).

Paraments	Models	Train Set		Test Set	
		R^2^_c_	RMSE_c_	R^2^_t_	RMSE_t_
Moisture content	RR	0.40	1.91	0.37	1.95
	RFR	0.41	1.54	0.40	1.65
Free amino acid content	RR	0.47	1.40	0.52	1.41
	RFR	0.66	0.23	0.65	0.33
Tea polyphenol content	RR	0.22	4.57	0.32	5.58
	RFR	0.24	1.12	0.46	1.67
Sugar content	RR	0.49	3.09	0.52	2.59
	RFR	0.34	0.12	0.70	0.13
Nitrogen content	RR	0.48	1.63	0.56	1.24
	RFR	0.92	1.49	0.50	1.51
Chlorophyll content	RR	0.78	0.97	0.78	0.94
	RFR	0.90	0.08	0.82	0.09

**Table 6 sensors-23-00571-t006:** Accuracy evaluation table of prediction results.

Paraments	Mean Absolute Error (MAE)	Root Mean Square Error (RMSE)	Determination Coefficient (R^2^ Score)
Moisture content	1.35	1.61	0.35
Free amino acid content	0.12	0.16	0.79
Tea polyphenol content	1.10	1.35	0.28
Sugar content	0.12	0.14	0.33
Nitrogen content	0.83	1.15	0.91
Chlorophyll content	0.01	0.02	0.97

## Data Availability

All data included in this study are available upon request by contact with the corresponding author.

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
