# Peer review of "Research on Rapid and Low-Cost Spectral Device for the Estimation of the Quality Attributes of Tea Tree Leaves"

_sensors, 2023, doi:10.3390/s23020571_

Round 1
Reviewer 1 Report
Review Report:
Journal Name and Manuscript ID: sensors-2117729
Title: Research on rapid and low-cost spectral device for detection of tea tree leaves quality attributes
Type of manuscript: Article
Authors: Jinghua Wang, Xiang Li, Wancheng Wang, Fan Wang *, Quancheng Liu, Lei Yan *
Submitted to Section: Multi-Sensor Information Fusion
Special Issue: ?
GENERAL EVALUATION:
The manuscript entitled as “Research on rapid and low-cost spectral device for detection of tea tree leaves quality attributes” deals with designing and testing of a prototype sensing system intended for estimating six ingredients of tea leaves (nitrogen, chlorophyll, free amino acid, moisture, polyphenol, sugar) in field conditions. The subject is important in terms of fast evaluation in the field conditions and eliminating costly and time-consuming lab analyses. There have been many research studies on the estimation of leaf chemical components by using various instruments (chromameter, spectroradiometer, chlorophyll meter, NDVI meter, machine vision, spectral imaging, NIR spectroscopy, etc.) There are also some studies which intended to develop low-cost practical prototype sensing systems for the evaluation of agricultural products such as weed, plant leaf chemistry, fruit quality, etc. However, these studies can be considered as in limited extend. Thus, I recommend acceptance after major revision of the manuscript for this high-impact factor journal. The subject of the manuscript fits the context of the Journal.
My main concerns to report to the Authors are given below:
1) The main question is that the sensor has two parts, a very small white reference surface on the left (diameter 38 mm) and sample area on the right (Figure 2). It looks like the sensor sees both sides at the same time. The sensor is in the middle, closer to the white reference. It is not explained how the sensor measures the data from both sides and distinguishes them. This is very critical point of the study. There are mainly two referencing methods; a) taking measurement from the white reference and then taking data from the sample in sequence b) using two sensors, one getting data from the intensity from the light source and the other acquiring data from the sample. Authors must clarify this design.
2) Second important question is about the sample collection.
- The authors must clarify how they managed to have samples with enough variations (variable rate fertilizer application?)
- Maintenance of trees: fertilizer, pesticide, and irrigation application?
- Did they apply any soil sampling and analysis?
- Did they collect all samples at the same time? Or in different occasions.
- Selection of leaves: young leaves or old leaves?
- Time of sampling: blooming stage, flowering stage?
- How many samples per one tree?
3) There are questions that must be clarified about the methods:
- Why did they select “Fuding white tea tree leaves”? Why not other types?
- How many leaves were placed inside the device. Did the leaves cover the surface completely?
- Did they use absorbance or reflectance in the modeling and testing (prediction)? They mention “absorbance transformation” (line 203)
- They eliminated some of the data (line 308-314) due to the effect of high humidity of the air in the tea plantation (from 264 to 220). Why did the humidity affect some samples but not others? Could this problem be related to sensor design?
- I think they used detached leaves. This should be mentioned in the paper.
- Did they use “full spectrum” or selected wavelengths? Did they use the wavelengths obtained from literature. They must provide this in a new table: which wavelengths were used for which leaf ingredients.
4) Figure 8 does not have a good meaning. Mean reflectance curves should be given for high, medium and low values for the most several important ingredients, for example nitrogen, etc.
5) Authors mention that the reflectivity in the 700~760nm was significantly elevated (Line 318; Figure 8). A comment should be added about the reason of this higher reflectance. Could it be related to a leaf ingredient?
6) The section in line 339-355 give information about the method and it should be moved to the Methods section probably to a place before line 299.
7) Some basic statistics should be given about the tea production in the world including the top five producer countries in the Intro section. Also, more information is needed about the tea types and their main differences. Some information is needed about the quality criteria of fresh tea evaluation for commercial purposes (during buying them from the farmers).
8) A better and short term must be used instead of "tree leaves ingredients information detection device". Maybe "leaf ingredient estimation sensor".
9) The device can predict some parameters with high accuracy (nitrogen, chlorophyll, free aminoacid) but some of them with lower accuracy (moisture, polyphenol, sugar) based on the R2 values. Please mention this in the Abstract section (near line 25).
10) They must explain how and for what purpose this system will be used by farmers / researchers / merchants in the field?
11) Discussion section must compare the results of the current study with the results of the previous similar studies. Were there similar prototype sensors? Are there differences between the performances? There are some studies in which prototype sensing system were developed. Some of them are given below for the consideration of the Authors:
- Habibullah M, MR Mohebian, R Soolanayakanahally, AN Bahar, S Vail, KA Wahid, A Dinh. Low-Cost Multispectral Sensor Array for Determining Leaf Nitrogen Status. Nitrogen, 2020, 1, 67–80; doi:10.3390/nitrogen1010007
- Maleki M, J. Massah, M. Dehghan. Application of a spectral sensor for the assessment of nitrogen content in lettuce plants. 2012, AJCS 6(2):188-193.
- Sekerli YE, M Keskin, Y Soysal. A low-cost prototype optical sensor to evaluate water, macro and micro elements of turfgrass clippings. Sensors and Actuators A: Physical, 2021, 323: 112615. https://doi.org/10.1016/j.sna.2021.112615
- Noguera, M.; Millan, B.; Andújar, J.M. New, Low-Cost, Hand-Held Multispectral Device for In-Field Fruit-Ripening Assessment. Agriculture 2023, 13, 4. https://doi.org/10.3390/agriculture13010004
- Duncan L, B Miller, C Shaw, R Graebner, ML Moretti, C Walter, J Selker, C Udell. Weed Warden: A low-cost weed detection device implemented with spectral triad sensor for agricultural applications. HardwareX. 2022. https://doi.org/10.1016/j.ohx.2022.e00303
12) Add some sentences to mention which ingredients were predicted with highest performance and which ones with lower performance, add R2 and RMSE values (Line 468) in Conclusion section. Comment why the prediction is lower for some ingredients.
13) All acronyms must have been spelled out the first time they are mentioned in the Abstract and other sections. All of them must be checked in the paper. Acronyms used in the tables and figures must be given in the figure itself or in the figure captions and under the tables.
14) In general, the writing style and the use of English language cannot be considered as satisfactory. There are lots of shortcomings. Manuscript needs significant revision in terms of English writing style and grammar. I included some of them below.
15) I listed some additional corrections, recommendations and questions below for the authors to improve the manuscript.
TITLE
Title should be modified. For example: "Research on rapid and low‐cost spectral device for the estimation of the quality attributes of tea tree leaves"
ABSTRACT
Line 14: Revise the sentence; too many nouns together? "tree leaves ingredients information detection device"
Line 14,25,30: Use "leaf ingredient" instead of "leaves ingredients"
Line 16: Delete the comma before "and"
Line 17,25: Use "leaf samples" instead of "leaves samples"
Line 20: Use "prediction performance" instead of "prediction effect"
Line 24: Give the numbers after each ingredient such as: moisture content (1.607 and 0.35)
Line 24: Give the numbers with two decimals such as 1.61 instead of 1.607
KEYWORDS
Line 30: Use "tea leaf ingredients" instead of "tea leaves ingredients"; Use "sensor design" instead of "device development"
INTRODUCTION
Line 38: Use "can be processed" instead of "can be made"
Line 39: Use "excitation" instead of "excitatory"
Line 47: Revise the sentence
Line 47,48,51: Use "chemical analysis" instead of "chemical experiments"
Line 48: Correct the typing error "a"
Line 67: Provide the full name for "QI"
Line 69: Phosphorus and potassium are not "trace elements", they are "macro elements", correct this.
Line 77: Correct the typing error "t"
Line 91-93: Do not repeat the names of the ingredients, they are already given in lines 88-89.
MATERIALS AND METHODS
Line 96: Add a main title before this line as "Materials"
Line 222: Change the title from "Materials and Methods" to "Methods"
Line 96,104,112: Use "tree leaf ingredients" instead of "tree leaves ingredients"
Line 105-107: Too many repeat for "unit design", please revise
Line 108: Explain why you used "Linux system, Qt software", is there a specific reason?
Line 116-117: Use "light spectrum" instead of "light band"
Line 120: Delete the comma before "and"
Line 126 Figure 2: Show the dimensions on the figure
Line 127 Figure 2: Use "the sample chamber unit" instead of "sample chamber unit"
Line 129: Did you use a single lamp or multiple lamps (beads), please clarify this
Line 129: Use "tungsten halogen lamp" instead of "halogen tungsten lamp"
Line 131: Use "bottom" instead of "in below"
Line 136: Give the number with one decimal (99.3%)
Line 138,139: Use "spectral data acquisition" instead of "spectral acquisition"
Line 146: Use "performance parameters" instead of "Performance Parameters", delete the dot
Line 148: This is an incomplete sentence, revise it.
Line 148,149: Write the numbers less than ten with letters, for example: two instead of 2
Line 153: Use "prediction performance" instead of "detection effect"
Line 154: This is not a separate sentence: "As shown in Figure 3", unite this with the previous sentence
Line 156 Figure 3: Use "the power supply" instead of "power supply"; Increase the size of the figure and the fonts; Provide the full name for VRM in the text or inside the figure; Use "DC fan" instead of "DC FAN"
Line 157,178: Delete the dot in the subtitle
Line 167: "where" instead of "Where"
Line 167-171: put the terms in the order same as in the formula
Line 168: Provide the full name for "CFM"
Line 176: Provide the meanings of all terms below the formula (Q, P, etc.)
Line 177: Use one decimal place, 6.0 instead of 6.00
Line 179: Combine the first two sentences to avoid the double usage of "RaspberryPi4 4B"
Line 180,216: Delete the comma before "and"
Line 190, Figure 4: Increase the size of the figure
Line 192: "process" instead of "Process"
Line 193: "a leaf ingredient" instead of "the ingredients parameter"
Line 194: "loads" instead of "loaded"
Line 201: "leaf sample" instead of "leaves sample"
Line 206, Figure 5: This is a flow chart, apply standard rules (boxes and arrows) and modify the figure, increase the size of the figure and the fonts, change the figure caption as "Flow chart of the software operation process", delete the dot at the end
Line 214, Figure 6: Increase the size of the figure and the fonts; Label each component in the figure; Provide a more meaningful caption for the figure instead of "Debugging and testing experiment"
Line 217-218: Revise the sentence
Line 219: Delete the comma
Line 220: Delete "measuring"
Line 222: Change the title from "Materials and Methods" to "Methods"
Line 224: Add "China" at the end
Line 231: Add comma before "respectively"
Line 233 Figure 7: Another photo of the tree from longer distance should be added
Line 237: Provide the approximate low temperature
Line 240,245,251,256: "where" instead of "Where"
Line 240-242: "mass" instead of "weight"; the unit of weight is "Newton" and the unit of mass is "g" or "kg"
Line 240, Formula 3: It should be: ((M1-M2)/M1); M0 is same as M1
Line 248: Add reference for the method
Line 245, Formula 4: Use parenthesis to clarify the terms in the equation, is it like this? (C/1000) x (V1/V2)
Line 245-247: Add the meaning and unit of "C"
Line 248: "the method of Lowenthal ‐ Neubaur" instead of "Iowenthal ‐ Neubaur"; Add reference; Explain the method briefly
Line 251, Formula 5: Add the units for the "Amount of extract" and "weight of sample"; Change "weight" to "mass"
Line 253: Add reference for the method
Line 254: "Kjeldahl method" instead of "Kjeldahl determination method"
Line 259: "sample quantity" instead of "sample quality"
Line 264: "Hoerl" instead of "HOERL"
Line 267: Delete the comma before "and"
Line 268, 282: "where" instead of "Where"
Line 268, Formula 7: "ridge" instead of "idge"
Line 270: "Add the meaning of "y"
Line 284: Check the typing of commas
Line 290: "the MAE" instead of "MAE"
Line 292: Explain the meanings of the terms under the formula
Line 293: "the RMSE" instead of "RMSE"
Line 295: Use a better definition for R2; it is related to the “explained variation” of the dependent variable by the prediction model
Line 296: Change as: "The closer the R2 to 1"
Line 298: Move this line before Line 292; also add the meaning of "y"
RESULTS
Line 301: Add an introductory sentence first; for example: "Some basic statistical data regarding the measured tea leaf ingredients are presented in Table 3."
Line 302-304: "the" instead of "The"
Line 306: Change the caption as: "Some basic statistical data on the measured tea leaf ingredients"
Line 306, Table 3: Add the number of data (n=264?); Put the minimum values first, then put the maximum values. Add standard deviation instead of variance.
Line 311,328,332,343,372: Delete the comma before "and"
Line 315, Figure 8: Increase the size of the figure and the fonts; Change the caption as "Original spectral curves of the tea tree leaf samples.
Line 318: "reflectivity in the" instead of "reflectivity of the"; Delete the dot before "and"
Line 338: "the tea tree leaf ingredients" instead of "tea tree leaves ingredients"
Line 340: Check the commas
Line 350: "verification analysis" instead of "verification experiment"
Line 348 and 350: "70% and 30%" was repeated
Line 356, Table 4: change as "ridge regression (RR) model and random forest (RFR) model; Use "RR" instead of "ridge" inside the table; Give all numbers with two decimals instead of three decimals inside the tables; Add horizontal lines after "sugar content" and "nitrogen content"
Line 357: "Table 4" instead of "Table 3"
Line 384, Figure 9: Increase the size of the figures and the fonts; Put two figures in a row; Delete the firts line of the title in each figure: "Comparison diagram of the measured and predicted values"
Line 385, Figure 10: Increase the size of the figures and the fonts; Put two figures in a row; Shorten the title of each figure by including only the ingredient name; Add R2 and RMSE values inside each figure on the top-left part
Line 390: Delete "device"
Line 390, Table 5: Give all numbers with two decimals instead of three decimals inside the tables
Line 391: "shows" instead of "showed that"
Line 391-402: Too many repetitions and very similar sentence structures; Delete and include more general comments
DISCUSSION
Line 406,447: Delete the comma before "and"
Line 420: "among" instead of "in"
Line 420: Add R2 value for chlorophyll content
Line 421: "at 700 nm" instead of "700 nm"
Line 424: Add R2 values for nitrogen and amino acid
Line 429: Revise the sentence; "obtained" was used twice
Line 438: Provide the numbers with two decimals
CONCLUSIONS
Line 457: Delete the comma before "and"
Line 459: "Display" instead of "page"
REFERENCES
- Use standard citing for all references.
- Include only the last names of the authors and the first letters of their first and middle names (Reference No:2,4,6,7,8,10,11,13,15,16,....etc. Check all others)
- Do not include the words "volume" and "page" (No:2,6,7,8,10,11,13,15, ... etc. Check all others)
- Words in the titles of some references start with capital letters (No:5,9,14, ... etc. Check all others)
- Use italic format for the plant names in Latin (No:7,... etc. Check all others)
- Do not use all capital letters in the author names (No:24,26,27,...etc.Check all others)
- Some pepers were listed twice: No 28,34
(end)
Author Response
Dear Reviewer:
Thanks for your comments concerning our manuscript entitled "Research on rapid and low‐cost spectral device for the estimation of the quality attributes of tea tree leaves". Those comments are all valuable and very helpful for revising and improving our paper, as well as the important guiding significance to our researches. We have studied comments carefully and have made a correction which we hope meets with approval. We tried our best to improve the manuscript and made some changes to the manuscript according to your suggestions. These changes are marked up using the “Track Changes” function, which will not influence the content and framework of the manuscript.
We appreciate for reviewer’s warm work earnestly and hope that the correction will meet with approval. The main corrections in the paper and the responses to the reviewer comments are in the attachment.
We hope this revision would meet with your permission. Thank you very much again.
Best wishes,
All authors

Reviewer 2 Report
In this paper, a tea tree leaves ingredients information detection device was developed based on multi-channel spectral sensor, which is conducive to real-time monitor the current quality of tea leaves. This work has a certain innovation. My major concerns are as follows, please make corrections.
Point 1: The language of the Abstract need to be embellished. In this part, the research process and results should be expounded systematically. In the meantime, the significance and purpose of this study should also be clarified here.
Point 2: line 93-95: ‘It was expected to provide an effective technical means for real-time monitoring the current quality of tea leaves, evaluating the status during tea trees growth, and im-proving the quality of tea production.’ Improve the wording as the text is confusing. Such as ‘real-time’ and ‘current’ are sort of repetitive.
Point 3: Present the objectives of the study at the end of Introduction section. If the general frame diagram of the research is added here, it will be easier for readers to understand this work.
Point 4: In line 126: It’s hard to identify the details in Figure 2.
Point 5: In line 223-232: Methodology for spectral data acquisition is a little confusing, such as what ‘one bud - one leaf and one bud -two leaves’ means?
Point 6: In Table 4: There seems to be some mistakes with the format of this table.
Point 7: RF means ‘Random Forest’, but what does RFR means? Please explain the meaning of the abbreviation.
Point 8: Authors have used a lot of imprecise words. These have terms could be translated through precise terms. Such as line 350, it should be ‘the sample data set for the device prototype verification experiment’ not ‘the sample data set of the device prototype verification experiment’.
Point 9: The authors should make the format of literatures listed in the references consistent.
Author Response

(The authors gave the same response as above.)

Round 2
Reviewer 1 Report
Review Report (Round 2):
Journal Name and Manuscript ID: sensors-2117729
Title: Research on rapid and low-cost spectral device for detection of tea tree leaves quality attributes
Type of manuscript: Article
Authors: Jinghua Wang, Xiang Li, Wancheng Wang, Fan Wang *, Quancheng Liu, Lei Yan *
Submitted to Section: Multi-Sensor Information Fusion
Special Issue: ?
Dear Editors,
I checked the revised manuscript. The authors accomplished most of the suggested changes and additions but there are some more to be revised. The manuscript is now relatively but nut fully in a better shape and content. There are still some minor changes and additions I suggest as listed below. I recommend acceptance of the paper after these minor revisions. There is no need for me to check it again.
List of Suggested Minor Revisions:
General comments:
- The Authors mention that they used “Track changes” property but I cannot see the deleted text and added text in the revised manuscript file.
- The most important and critical point is still the same subject. They say that they modified Figure 2 mentioning that they measure the reflectance from the white reference panel first and then from the leaf sample. They modified Figure 2 but they still use the old version of Figure 1 (white reference and tea leaf sample are shown as side by side), Figure 1 must be modified as well. They have to mention the sequential measurement (first white panel and then leaf sample) in the flow chart in Figure 5 too.
- What is the size of the white reference panel? Is it 38mm as mentioned in Line 140. I think this is too small. Are they sure that this size was enough? (I do not think this would be enough?).
- They mention in the “Author Response” file that the white reference panel is visible in the figure but it is not visible in Figure 2 but visible in Figure 1 (old version).
- The Authors added a new table (Table 1) given in the “Author Response” file but this table was not added to the revised paper.
- The Authors mention that “we collected both young leaves or old leaves when sampling” in the “Author Response” file (page 2). I think this is not right. In leaf sampling and analysis, usually the leaves that completed full maturity (not too young - not too old) are collected and analyzed. The authors must give a reference for their method.
Some minor but important revisions are also recommended below:
Line 17,54,59: Delete one of the repeated words “leafleaf”
Line 35: Add the name of tea in “Latin”
Line 36-37: Add a reference
Line 39: Use a better term instead of “mature drink”
Line 85: “tree leaf” instead of “treeleaf”
Line 118: “leaf ingredient” instead of “leaf ingredientsleaf ingredient”
Line 289: Put comma between the chemical bond names
Figure 4: Please increase the size of the figure and the fonts
Figure 7: Provide which line represents high, medium and low nitrogen contents. Add label “a” and “b” on the figure. Increase the font size of the text in the figure.
Figure 8: Increase the font size of the text in the figure.
Figure 9: Increase the font size of the text in the figure.
Table 5: Provide the data in the table with two decimal places
Line 493-602: The formats of the references are not standard. Start with last name of the first author.
(end)
